# Strain in the Midbrain: Impact of Traumatic Brain Injury on the Central Serotonin System

**DOI:** 10.3390/brainsci14010051

**Published:** 2024-01-05

**Authors:** Christopher J. O’Connell, Ryan S. Brown, Taylor M. Peach, Owen D. Traubert, Hana C. Schwierling, Gabrielle A. Notorgiacomo, Matthew J. Robson

**Affiliations:** 1Division of Pharmaceutical Sciences, James L. Winkle College of Pharmacy, University of Cincinnati, 231 Albert Sabin Way, Cincinnati, OH 45267, USA; oconnech@mail.uc.edu (C.J.O.); brown5rs@mail.uc.edu (R.S.B.); peacht2@mymail.nku.edu (T.M.P.);; 2Department of Biomedical Engineering, Pratt School of Engineering, Duke University, Durham, NC 27708, USA; owen.traubert@duke.edu; 3College of Medicine, University of Cincinnati, Cincinnati, OH 45267, USA; notorgga@mail.uc.edu; 4Neuroscience Graduate Program, College of Medicine, University of Cincinnati, Cincinnati, OH 45267, USA

**Keywords:** traumatic brain injury, serotonin, drug discovery

## Abstract

Traumatic brain injury (TBI) is a pervasive public health crisis that severely impacts the quality of life of affected individuals. Like peripheral forms of trauma, TBI results from extraordinarily heterogeneous environmental forces being imparted on the cranial space, resulting in heterogeneous disease pathologies. This has made therapies for TBI notoriously difficult to develop, and currently, there are no FDA-approved pharmacotherapies specifically for the acute or chronic treatment of TBI. TBI is associated with changes in cognition and can precipitate the onset of debilitating psychiatric disorders like major depressive disorder (MDD), generalized anxiety disorder (GAD), and post-traumatic stress disorder (PTSD). Complicating these effects of TBI, FDA-approved pharmacotherapies utilized to treat these disorders often fail to reach the desired level of efficacy in the context of neurotrauma. Although a complicated association, decades of work have linked central serotonin (5-HT) neurotransmission as being involved in the etiology of a myriad of neuropsychiatric disorders, including MDD and GAD. 5-HT is a biogenic monoamine neurotransmitter that is highly conserved across scales of biology. Though the majority of 5-HT is isolated to peripheral sites such as the gastrointestinal (GI) tract, 5-HT neurotransmission within the CNS exerts exquisite control over diverse biological functions, including sleep, appetite and respiration, while simultaneously establishing normal mood, perception, and attention. Although several key studies have begun to elucidate how various forms of neurotrauma impact central 5-HT neurotransmission, a full determination of precisely how TBI disrupts the highly regulated dynamics of 5-HT neuron function and/or 5-HT neurotransmission has yet to be conceptually or experimentally resolved. The purpose of the current review is, therefore, to integrate the disparate bodies of 5-HT and TBI research and synthesize insight into how new combinatorial research regarding 5-HT neurotransmission and TBI may offer an informed perspective into the nature of TBI-induced neuropsychiatric complications.

## 1. Introduction

Within this review, we will firstly seek to provide researchers with a relevant perspective on the intersection of modern serotonin (5-HT) and traumatic brain injury (TBI) research domains. Secondly, we will discuss the efficacy and use of selective serotonin reuptake inhibitors (SSRIs) in the context of TBI-elicited psychiatric disturbances. Finally, we will integrate the seminal studies that have functionally, anatomically, and genetically mapped the murine central 5-HT system and discuss the implications of these results for potential therapeutic development in humans afflicted with TBI-induced psychiatric disturbances. The research domains encompassing preclinical and clinical 5-HT and traumatic brain injury research are vast, and as such, we acknowledge that not all of the relevant literature is reviewed and discussed herein. Additionally, it is not within the scope of this review to make statements or predictions using data derived from the referenced manuscripts; rather, we aim to conceptually synthesize the conclusions presented in the referenced manuscripts and present the reader with a synopsis of the current state of both the 5-HT and TBI research fields. Further, our review is not a meta-analysis of currently available data, nor is it intended to make recommendations regarding treatments or the efficacy of currently available treatment modalities. It is solely our intent herein to integrate the seemingly disparate fields of 5-HT and TBI research. As such, we recognize that the format of this review is not in line with classic PRISMA guidelines; however, we have aligned our review as much as possible with applicable PRISMA guidelines where at all feasible. 

TBI is a pervasive and untenable burden to public health and wellness and comprises a significant proportion of accidental deaths and disabilities in the United States of America [1]. The Centers for Disease Control and Prevention (CDC) estimates that greater than 2.5 million people are affected by TBI annually [2], though this figure is considered by many to be a gross underrepresentation of the true prevalence of the disease. The health care burden of TBI has a significant economic impact that extends far beyond immediate treatment expenses, with nearly USD 77 billion spent annually on acute, protracted, and extenuating health care costs associated with the various forms of neurotrauma [3]. 

The etiology of clinical TBI is extraordinarily heterogenous, and individuals who have experienced similar environmental exposures often present to the clinic with markedly different injury pathologies and subsequently go on to have disparate outcomes [4]. Although often presenting distinctly in the clinic, heterogenous TBI cases are generally stratified by their severity. The classifications of mild (mTBI/concussion), moderate, and severe TBI were established following the publication of the Glasgow Coma Scale (GCS) in 1974, which is a widely implemented guideline that categorizes neurotrauma by assessing motor and verbal responses to prompting, eye movement, and other metrics [5]. All forms of closed-head neurotrauma are known to catalyze enduring perturbations in the function of the central nervous system (CNS), and the consequences of TBI may include psychiatric complications, sleep disturbances, fatigue, and persistent, intractable headache [6]. Presently, there are no FDA-approved pharmacotherapies available specifically for the acute or chronic treatment of TBI symptoms, and the current therapeutic strategies employed to attenuate enduring psychosocial complications are insufficient. 

Although the precise molecular mechanisms associated with psychosocial complications following TBI have yet to be conclusively established, the gross physical effects of mTBI on the brain following various forms of injury have been comprehensively documented. The transmission of mechanical stress to brain tissue, denoted the primary injury phase of TBI, occurs during injury induction, wherein the brain tissue can experience heterogenous rotational and linear acceleration/deceleration inertial forces that impart significant shearing and torsional stress to the brain parenchyma [7,8,9]. Axonal projections are particularly susceptible to damage from inertial forces given their long filamentous morphology, and axonal shearing greatly increases the probability of unconsciousness following closed-head injury [10,11]. The average neuronal density within the brain varies with depth, lending to the differential distribution of inertial forces during impulsive head kinematic events [12]. Kinematic forces imparted by inertial loading of the tissue may displace brain tissue within the rigid body of the skull, and this rapid transmission of mechanical energy to the brain parenchyma can induce physical trauma to the tissue [9]. This process of inertial loading is a defining feature of mTBI and can even occur in the absence of a direct impact, induced exclusively by impulsive head kinematics [7,13].

The transmittance of external, environmental kinematic forces to brain tissue can also vary across separate, disparate brain regions due to the inherent, non-uniformity of cell types and densities comprising brain tissue [14]. Notably, subcortical regions such as the midbrain are preferentially impacted by head impact and blast forces. Repetitive head impacts (RHI) progressively reduce the structural integrity of the white matter within the midbrain (i.e., axonal tract damage) [15], and TBI results in significant midbrain diffuse axonal injury that correlates to states of unconsciousness [13]. Further, clinical imaging studies and post-mortem analyses of affected individuals have corroborated preclinical observations denoting the midbrain as being preferentially impacted by RHI and mTBI [16,17], making the midbrain and the cellular populations that comprise the midbrain a relevant research target. Cumulatively, these clinical studies indicate that the dorsal zone of the midbrain raphe nuclei, a region with diverse serotonergic neuron populations, is preferentially impacted by mild TBI, with implications for the development of mood disturbances like depression following injury induction. 

Structural and metabolic changes associated with chronic traumatic encephalopathy (CTE) and mTBI have been quantified in clinical settings employing [F-18] FDDNP, an imaging agent for fibrillar insoluble protein aggregates, and positron emission tomography (PET) imaging to establish topographic brain localization of neuropathology in areas involved in mood disorders in individuals with mTBI [16]. Among the observed [F-18] FDDNP+ signal patterns in individuals with mTBI compared to naïve participants, pattern T1 was defined as predominantly subcortical in brainstem (midbrain) with localized involvement of the limbic medial temporal lobe structures (limited to amygdala) [16]. Involvement of the midbrain and amygdala is corroborated by neuropathological examinations of deceased American football players who reported premortem functional impairments. The acceleration and deceleration of the brain parenchyma in the sagittal plane during impulsive head kinematic events impart physical strain on the midbrain tissue, which is correlated with positive [F-18] FDDNP PET results [16]. While the principal strain induced within the midbrain areas during the induction of concussions cannot be recorded or quantified in premortem individuals, the effects of linear acceleration in the sagittal plane may be recapitulated computationally. 

Specifically, physical interactions between the parenchyma of the human brain and the environment that occur during acceleration/deceleration events associated with whiplash can be computationally constructed using a human head finite element modeling system [17]. Though diffusion tensor imaging (DTI) is the most sensitive imaging modality for visualizing minute damage to white matter tracts, up to half of patients with head injuries do not have positive DTI results measured by changes in fractional anisotropy and mean diffusivity [17,18]. To circumvent the limitations of modern imaging techniques, the Total Human Model for Safety (v.5) 50th percentile adult male model was utilized to simulate acceleration/deceleration events associated with car crashes at various speeds [17]. At all speeds simulated, the dorsal zone of the raphe nuclei demonstrated comparatively higher principal shearing strain values than other brain regions, providing evidence for midbrain vulnerability to damage induced by the transmission of inertial forces. ^18^F-fluorodeoxyglucose positron emission tomography was employed in individuals with chronic cognitive symptoms and no visible lesions following TBI to quantify glucose metabolism. Importantly, glucose metabolism is significantly decreased across the entire bilateral prefrontal areas, brain regions previously associated with depression following TBI [19], with compensatory increases in glucose metabolism in the limbic systems and cerebellum detected in patients with mTBI [20]. Incredibly, the reported pattern of altered metabolism in patients with mTBI was topographically similar to patterns of metabolism reported in non-injured depressed patients [17]. 

Though the mechanisms underlying TBI-elicited neuropsychiatric disturbances have not been experimentally resolved, ongoing preclinical and clinical experimental paradigms interrogating the midbrain have identified a rich landscape of neurons that produce 5-HT, expansively project to disparate regions of the brain, and exert global influence over a plethora of behavioral and physiologic phenotypes [21]. As depicted in Figure 1. the molecular effects of TBI on populations of brainstem serotonergic neurons and elicits long-term neurological complications remains a domain of intense scientific inquiry, as the delineation and characterization of these neurons is simultaneously ongoing. 

## 2. Materials and Methods

### Search Strategies

A comprehensive, computer-based search was conducted, and studies were identified through electronic queries of database resources including PubMed and bioRxiv. Articles were screened with no language restriction imposed. Databases were searched for terms related to 5-HT and traumatic brain injuries, neurological manifestations of mood disturbances in humans, preclinical modeling of traumatic brain injuries, and preclinical interrogation of the 5-HT system. The keywords that could be contained in either the title or abstract were as follows: “traumatic brain injury”, “traumatic brain injury AND serotonin”, “scRNA-seq AND serotonin AND elife[jour]”, “scRNA-seq AND serotonin AND Pet1 AND elife[jour]”, “traumatic brain injury AND depression”, “traumatic brain injury AND systematic review”, “serotonin AND systematic review”, “traumatic brain injury AND metabolism”, “serotonin AND metabolism” “serotonin AND receptor AND pharmacology”, “traumatic brain injury AND serotonin AND receptor AND pharmacology”, “serotonin AND behavior”, “traumatic brain injury AND serotonin AND behavior”, “traumatic brain injury AND behavior”, “raphe nucleus AND serotonin”, “dorsal raphe nucleus AND neuron”, “dorsal raphe nucleus AND serotonin AND neuron”, “traumatic brain injury AND positron emission tomography”, “traumatic brain injury AND finite element modeling AND serotonin”, “traumatic brain injury AND selective serotonin reuptake inhibitors”, “traumatic brain injury AND pain”, “traumatic brain injury AND serotonin AND pain”.

## 3. Traumatic Brain Injury and Psychiatric Disease

### TBI and Mental Health Statistics

After sustaining a head injury, the quality of life for affected individuals may become substantially diminished in both the immediate and long term. In individuals who live with protracted negative outcomes, the disabilities that are associated with poor quality of life have been primarily attributed to neurobehavioral features [19,22]. Various facets of normal day-to-day functioning may be affected, including difficulty attending to social relationships, finding gainful employment, and participating in broader social capacities [19,23,24,25]. Intractable, treatment-resistant depression and other serious psychiatric disturbances are common among individuals who have sustained neurotrauma [6,19,26]. Among these individuals, the development of MDD occurs at a prevalence rate of greater than 25%, with some reports suspecting the prevalence may be as high as 50% measured using the Patient Health Questionnaire (PHQ) depression and anxiety modules [19,27,28,29]. A single prospective study that tracked the psychological outcomes of individuals who had been submitted to a hospital following a head injury found that 52% of the individuals developed MDD within one year following the injury [30]. More broadly, it is suspected that the prevalence of MDD in TBI patients who sustained mild, moderate, or severe neurotrauma increases from 33% to 42% in the first year to over 60% within 7 years, as measured using the DSM-IV criteria, Patient Health Questionnaire, International Classification of Diseases—Ninth Revision [19,23,31,32,33,34].

## 4. The 5-HT System and TBI

### 4.1. History of 5-HT in Traumatic Brain Injuries

5-HT is a biogenic monoamine first discovered by Vialli and Erspamer in enterochromaffin cells of the gut in 1937 [35]. Following its discovery in 1937, 5-HT was identified throughout the human anatomy and was found to act to modulate peripheral physiology, including the vascular, cardiac, pulmonary, endocrine, and gastrointestinal systems [36,37,38,39,40,41,42,43,44]. Though the majority of 5-HT resides in the periphery where approx. 90% of it is synthesized, 5-HT is also present in the CNS and is produced by serotonergic neuron populations that originate in the midbrain and brainstem raphe nuclei (RN), project to disparate brain regions, and influence physiologic and behavioral phenotypes [21]. In humans, 5-HT neurons are stratified in the brainstem and given the designation B1–B9 dependent on their precise anatomical localization and projection region [45]. Neurons associated with groups B1–B3 are localized in the hindbrain medulla and descend to innervate the lower medulla and upper spinal column, whereas neurons in groups B4–B7 are localized anterior to the pons and midbrain, and projections from these groups innervate the midbrain and forebrain [46]. In the midbrain, serotonergic neurons are further delineated anatomically into the dorsal and median zones of the raphe nuclei (DR and MR, respectively), which participate in expansive circuits that regulate many mental processes [46,47,48]. In fact, serotonergic neuron architectures are so expansive in the CNS that nearly every brain region expresses receptors for 5-HT in a receptor subtype-specific manner, even though fewer than one in every million neurons is capable of producing 5-HT [49]. When considering the scarcity of total serotonergic neurons in contrast to the vast diversity of physiologic and psychological phenomena regulated by 5-HT signaling, a depiction of a heterogenous neuronal system with precisely tuned functions and structures emerges. There are currently 15 known genes that encode the 5-HT receptor family (5-HT_1_–5-HT_7_) with a total of 13 G-protein-coupled receptors and a family of ligand-gated ion channels [50,51]. Additionally, whether mood disturbances and other psychosocial features of TBI can emerge exclusively as a result of altered 5-HT signaling through these receptor families, or through distinct molecular mechanisms, is not currently known. Though historical and ongoing postmortem, imaging, and computational studies have unilaterally demonstrated that the midbrain RN are specifically vulnerable to the inertial forces associated with mTBI, these insights have not been successfully translated into mechanism-derived therapeutic strategies that target the 5-HT system.

### 4.2. 5-HT Metabolism and TBI

Though the molecular features that govern basal function of 5-HT signaling have been resistant to TBI-specific drug discovery efforts given that their activity is mediated through complex mechanisms like signaling through multiple receptor families, the enzymatic process of 5-HT synthesis within serotonergic neurons has been readily characterized. As depicted in Figure 2, 5-HT is a tryptophan product that is converted from L-Tryptophan to 5-Hydroxytryptophan by tryptophan hydroxylase 1 and 2 (Tph1/2), which are members of a superfamily of aromatic amino acid hydroxylases and represent the rate limiting step in 5-HT synthesis [52]. 5-Hydroxytryptophan is metabolized to 5-HT by aromatic l-amino acid decarboxylase (AADC), which can be catabolized via multiple pathways into 5-hydroxyindolacetic acid (5HIAA), melatonin (5-methoxy-*N*-acetyltryptamine), and other biologically active molecules [53].

TBI-elicited changes to the process of 5-HT synthesis or changes in total levels of 5-HT in the brain, by way of altered metabolic programs, may be relevant to delineating the generation of abnormal behaviors, as current FDA-approved therapies to treat disorders like MDD and anxiety are thought to exert their effects by altering the abundance of 5-HT levels in the synapse [54,55]. While research paradigms have depicted changes in cellular metabolism following mTBI in clinical populations [17] and in preclinical studies [56], the effects of neurotrauma on the metabolism of 5-HT and 5-HT-derived products have not been fully delineated in humans; however, they have been examined in a limited number of preclinical rodent studies. Preclinical TBI paradigms alter tryptophan metabolism, with unknown implications for the generation of 5-HT and other tryptophan-derived metabolic products [57]. As the connection between altered 5-HT signaling and changes in cellular metabolic activity has been established in both preclinical and clinical studies of traumatic brain injuries, additional work has sought to quantify changes in the gross metabolism of 5-HT specifically and the impact of altered cellular metabolism on the 5-HT system. In a preclinical model of mTBI in rats, TBI-elicited increases in the total levels of positive 5-HT immunohistology present in the brainstem were reported [58], with a separate study identifying injury-elicited increases in the abundance of *Tph2* mRNA transcripts in the midbrain dorsal raphe nucleus (DRN) following a blast exposure [59].

Parallel to the observed aberrant glucose metabolism detected in the cortex of the injured brain in humans [17], specific metabolic features of TBI that implicate the 5-HT system have been identified across various preclinical models for TBI. In a rodent model of brain injury elicited by focal freezing lesions of the cortex, widespread decreases in glucose utilization and increases in 5-HT metabolism were detected in the injured hemisphere and subsequently attenuated by inhibiting the synthesis of 5-HT [60]. Further, autoradiographic paradigms were used to measure the accumulation of ^14^C-labeled 5-HT analogue α-methylserotonin and quantify changes in the rate of 5-HT synthesis. Three days post-injury (DPI) corresponded to both the period of lowest glucose utilization and the period of greatest 5-HT abundance in the cortex and dorsal hippocampus. Additionally, 3 DPI corresponded to significant increases in 5-HT synthesis in the DRN, which did not extend to the median raphe nucleus though serotonergic neurons are present in both regions [60]. Together, these studies have yielded significant evidence for the ability of various forms of TBI to dramatically alter the landscape of 5-HT metabolism in the midbrain and brainstem, which subsequently impacts other CNS metabolic functions through altered serotonergic signaling.

### 4.3. 5-HT Receptor Pharmacology in TBI-Elicited Behavioral and Nociceptive Pathologies

The impact of the large family of 5-HT receptors on therapeutic development for TBI is two-fold; the great abundance of distinct 5-HT receptor subtypes affords an increased number of potential therapeutic targets while simultaneously increasing the probability of non-specific drug–target interactions and subsequent off-target pharmacological effects. Despite this, pharmacologically targeting the families of various 5-HT receptors in a receptor-specific manner has yielded therapies for other disease states and disorders [61] and, therefore, may also provide valuable drug discovery targets in the context of TBI.

As depicted in Table 1, a significant preclinical research effort has sought to evaluate 5-HT receptor pharmacology in the context of TBI-elicted behavioral disruptions. Pharmacological modulation of the 5-HT_1A_ receptor has been routinely examined, as extensive historical work has successfully delineated the structure and function of the receptor and has localized the receptor to diverse brain regions including the hippocampus, forebrain, and DRN [62] through PET studies using [^11^C]-WAY100635 to localize the 5-HT_1A_ to these brain regions in humans [63]. In these regions, 5-HT signaling transduced through the 5-HT_1A_ receptor is implicated in memory formation, cognition, and thermoregulation [62,64]. The functional integrity of the 5-HT_1A_ receptor and the basal cognitive processes that emerge from 5-HT_1A_ activity are reportedly affected by various CNS injuries, including TBI, indicating that the receptor may be a relevant therapeutic target [64,65]. Pharmacologic manipulation of the 5-HT_1A_ receptor has been extensively investigated in preclinical models of TBI. Significant experimental evidence demonstrates that activation of the 5-HT_1A_ receptor through the administration of selective agonists, including 8-OH-DPAT, increases performance in various memory and cognition tasks following CNS injury while simultaneously reducing the deleterious effects of injury on neuron function, abundance, and morphology [64,66,67,68]. A preclinical study that utilized a combinatorial approach of environmental enrichment and administration of 5-HT_1A_ receptor agonist buspirone following TBI reported enhanced spatial learning in rats [69]. Additionally, a singular clinical study that involved progressive intravenous administration of the 5-HT_1A_ agonist BAYx3702 (Repinotan) in individuals who sustained a severe TBI found that the Repinotan-treated patients had a greater proportion of good or moderate outcomes instead of severe disability compared to the placebo group [70]. Though 5-HT_1A_ activation through a plethora of agonists has demonstrated preclinical and clinical promise in the context of TBI, only buspirone is utilized as a treatment for psychiatric disturbances. As such, therapies targeting the activation of the well-characterized 5-HT_1A_ receptor may be highly amenable to translational research paradigms and merit consideration as a promising strategy to attenuate the cognitive symptoms of TBI.

The literature delineating the roles of the other 5-HT receptors in the 5-HT_1_ family in neurotrauma is scant, with little evidence suggesting a beneficial or deleterious role for 5-HT_1B,1D,1E_ or 5-HT_1F_ receptor modulation in TBI. Though neither the effects of neurotrauma on the 5-HT_1B–1F_ receptors nor the impact of 5-HT_1B–1F_ receptor modulation on the outcomes of TBI have been comprehensively studied, compounds that target other members of the 5-HT_1_ receptor subfamily have been used preclinically with various effects on animal behavior [62]. For example, triptans are agonists of the 5-HT_1B_ and 5-HT_1D_ receptors and are utilized clinically to resolve the symptoms of migraines, which may spontaneously occur after TBI and are often intractable without the use of these medications [71]. The clinical and experimental applications of drugs like triptans that modulates the family of 5-HT receptors in humans is reported in Table 2.

The 5-HT_2_ family of receptors, which include the 5-HT_2A,2B_ and 5-HT_2C_ receptor subtypes, have been implicated in perception and mood, and, therefore, represent promising targets for pharmacologically ameliorating the perceptual and emotive disturbances associated with TBI and other psychiatric disorders such as MDD and PTSD [72,73,74,75]. Agonism of the 5-HT_2_ receptor family is also likely to be involved in generating hallucinations, as there is a significant correlation between hallucinogenic properties of 5-HT_2_ receptor agonists and their respective 5-HT_2_ receptor binding affinities [62,76]. Specifically, pharmacologic manipulation of the 5-HT_2A_ receptor by agonists like lysergic acid diethylamide (LSD) and mescaline elicit potent hallucinogenic effects while simultaneously inducing euphoria [77]. Additionally, compounds that modulate 5-HT_2A_ have been rigorously examined in the context of disorders that affect mood, perception, and memory like MDD [78], but comparatively little has been done to characterize the capacity of 5-HT_2A_ agonists to attenuate TBI-elicited neuropsychiatric disturbances. In a singular study that utilized a mild, closed-head model for murine TBI, cortical 5-HT_2A_ receptor ligand binding affinities were increased, eliciting decreases in normal social behaviors 10 days following injury. These changes were normalized through progressive dosing of 5-HT_2A_ agonist 2,5-dimethoxy-4-iodoamphetamine (DOI) [79]. In parallel with these preclinical findings, elevated levels of the 5-HT_2A_ receptor were found in the orbital-frontal cortex of antisocial humans, suggesting that changes in the abundance or function of cortical 5-HT_2A_ receptors play a role in generating aberrant psychosocial phenomena in general, and potentially in the context of neurotrauma [80]. Conversely, there are little data available depicting the effects of 5-HT_2B_ receptor modulation on behavioral phenotypes and no data examining the 5-HT_2B_ receptor in TBI, though a singular preclinical study found that the 5-HT_2B_ receptor is required for the 5-HT-selective antidepressant actions of the FDA-approved therapies for MDD [81].

The 5-HT_2C_ receptor, like the 5-HT_2A_ receptor, is implicated in the generation of hallucinogenic experiences, with agonists of the 5-HT_2C_ receptor inducing schizophrenia-like behaviors in mice [82]. While there are no studies explicitly examining the role of the 5-HT_2C_ receptor in the context of TBI-elicited behavioral disturbances, there is a single preclinical study reporting that agonism of the 5-HT_2C_ receptor by agomelatine improved rodent coordination following craniotomy-induced brain injuries [83]. The scarcity of preclinical and clinical behavioral studies may be a result of the complex pharmacology of the 5-HT_2C_ receptor, as activation of the receptor is reported to be anxiogenic in some studies, yet may demonstrate therapeutic potential in treating the symptoms of depression and obsessive-compulsive disorder (OCD) in others [62,84]. Though there are no studies explicitly examining the role of the 5-HT_2C_ receptor in generating anxiety phenotypes or attenuating MDD or OCD symptomology following neurotrauma, TBI unilaterally increases the risk for developing these disorders, potentially illustrating a role for the 5-HT_2C_ receptor in drug discovery efforts.

Unfortunately, there are a paucity of studies connecting the remaining 5-HT receptor families and TBI-elicited psychiatric disturbances, as basic receptor characterization and drug development efforts encompassing the 5-HT_3–7_ receptor families are ongoing even in the absence of neurotrauma. While a singular study illustrated a beneficial effect of concomitant 5-HT_1A_ and 5-HT_7_ receptor activation on behavioral phenotypes following TBI, more extensive research is required before the 5-HT_3–7_ receptor families may be implicated in driving TBI-induced psychosocial sequelae [68].

There is also significant evidence for the role of 5-HT pharmacology in modulating pain, as activity of the 5-HT system either facilitates or inhibits the pain response through 5-HT signaling at various receptor subtypes [85]. In the context of chronic pain, 5-HT signaling acts in a nociceptive, antinociceptive, or biphasic manner depending on the hodology, receptor subtype, and anatomical localization [86]. Though there are a paucity of studies examining the roles of the 5-HT_3–7_ receptor families in the context of TBI-elicited neuropsychiatric disturbances, preclinical interrogation of these receptors has implicated signaling through spinal 5-HT_3_ and 5-HT_7_ receptors in the generation of pro- and antinociception, respectively, using the formalin test, carrageenan model and intrathecal injections of 5-HT_3_ antagonists and 5-HT_7_ agonists in Sprague-Dawley rats [87]. At the supraspinal level, neuronal circuitry originating in the periaqueductal grey (PAG), a midbrain structure, underlies regulation of the descending pathway in humans, with biphasic activity of the descending PAG circuit being an integral component of the endogenous analgesic system [88]. In the context of TBI, intrathecal administration of 5-HT_3_ receptor antagonist ondansetron following controlled cortical impact in C57Bl/6 mice attenuated nociceptive sensitization in a dose-dependent manner [89].

**Table 1 brainsci-14-00051-t001:** 5-HT receptor family pharmacology and relevance to preclinical TBI-related behavioral disruptions.

Receptor	Species	Study Design	Model|Severity	Pharmacology	Behavioral Outcomes|Measured Effects
5-HT_1A_	Mouse, NMRI	Male (10–11 weeks) *N* = 10/group	Mild TBI Weight-drop model	Agonism, 8-OH-DPAT Antagonism, WAY-100635	Subthreshold administration of 8-OH-DPAT improved performance in forced swim, sucrose preference, and tail suspension test, whereas 5-HT_1A_ blockade increased depression-like phenotypes [90].
5-HT_1B_	Mouse, C57Bl/6	Male (8 weeks) *N* = 6–12/group	Repetitive mild TBI CHIMERA model	Androgenic anabolic steroids	Receptor-specific pharmacology not evaluated in the context of TBI-elicited behavioral changes, though receptor expression was reduced in response to anabolic steroids irrespective of TBI treatment [91].
5-HT_1D_	Rat, Sprague-Dawley	Male, *N* = 128	Mild TBI Weight-drop model	Agonism, Triptans	Acute sumatriptan treatment attenuated cephalic tactile hypersensitivity at 72 h post-injury and elicited conditioned place preference in injured subjects 7 days post injury [92].
5-HT_2A_	Mouse, C57Bl/6	Male (9–16 weeks) *N* = 10–12/group	Mild TBI Blast model of mTBI	Agonism, DOI Antagonism, M100907	Decreases in social preference and social dominance phenotypes measured by 3-chamber test and tube test, respectively, were attenuated by progressive DOI administration post injury [79].
5-HT_2B_	Mouse, transgenic	Male (7–9 weeks) *N* = 3–14/group	-	Agonism, Fluoxetine Antagonism, RS127445	Not evaluated in the context of TBI-elicited behavioral changes, though receptor activation may be required for antidepressant mechanisms of SSRIs in MDD [81].
5-HT_2C_	Rat, Sprague Dawley	Male and Female *N* = 90	Surgical craniotomy	Agonism, Agomelatine	Not evaluated in the context of TBI-elicited behavioral changes, though receptor activation acutely reduced locomotor deficits following surgical craniotomy [83].
5-HT_3_	Rat, Sprague Dawley	Male *N* = 6–8/group	Formalin and carrageenan pain model	Antagonism, SB-269970	Not evaluated in the context of TBI-elicited behavioral changes, though receptor blockade may attenuate nociception associated with neurotrauma [87].
5-HT_4_	Mouse, C57Bl/6	Male, (7–8 weeks) *N* = 7–8/group	-	Agonism, Prucalopride and velusetrag	Not evaluated in the context of TBI-elicited behavioral changes, though receptor activation by agonists prucalopride and velusetrag improved facilitation of contextual fear extinction in a preclinical model of Parkinson’s disease [93].
5-HT_5A_	Rat, Wistar	Female (6–7 weeks) *N* = 6/group	Nerve ligation pain model	Antagonism, Methiothepin and SB-69951	Not evaluated in the context of TBI-elicited behavioral changes, though receptor blockade reduced neuropathic antiallodynic effect of 5-HT dosing [94].
5-HT_6_	Rat, Sprague Dawley	Male (6–7 weeks) *N* = 92	Moderate TBI Controlled cortical impact model	Agonism, WAY-181187	Impairments in spatial reference memory measured by the Morris water maze were alleviated following WAY-181187 administration [95]
5-HT_7_	Rat, Sprague Dawley	Male *N* = 6–8/group	Formalin and carrageenan pain model	Agonism, AS-19	Not evaluated in the context of TBI-elicited behavioral changes, though receptor activation may attenuate nociception associated with neurotrauma [87].

**Table 2 brainsci-14-00051-t002:** 5-HT receptor family pharmacology and relevance to TBI-related behavioral disruptions or other neurological disorders in humans.

Receptor	Study Design	Condition|Disorder	Pharmacology	Behavioral Outcomes|Pharmacological Effects
5-HT_1A_	Blinded 60 patients enrolled	Severe TBI	Agonism, Repinotan	Proportion of patients having good outcome was slightly greater in repinotan-treated patients (60%) than in controls (50%) measured by the Glasgow coma scale [70]. A greater proportion of severe TBI patients who received repinotan treatment scored more highly on measures of responsivity to commands, including eye opening, orientated verbal responses to questions, and obeying motor movement commands.
5-HT_1B_	Male and Female 100 individuals	Mild TBI Post-traumatic headache	Agonism, Triptans	Not evaluated in the context of TBI-elicited behavioral changes, though receptor agonism may mitigate post-traumatic headaches [96]. Seventy percent of individuals treated with triptan-lass medications experienced reliable headache relief compared to 42% that were administered alternative headache-abortive medications.
5-HT_1D_	Male and Female 100 individuals	Mild TBI Post-traumatic headache	Agonism, Triptans	Not evaluated in the context of TBI-elicited behavioral changes, though receptor agonism may mitigate post-traumatic headaches [96]. Seventy percent of individuals treated with triptan-class medications experienced reliable headache relief compared to 42% that were administered alternative headache-abortive medications.
5-HT_2A_	Male Middle aged 35 patients enrolled	Mild to Moderate TBI	-	Not evaluated in the context of TBI-elicited behavioral changes, though 2/3rds of TBI patients harbored 5-HT_2A_ autoantibodies, the concentrations of which were correlated with other neurological comorbidities, including Parkinsons’ disease, other dementias, and painful neuropathies [97].
5-HT_2B_	-	Migraine	Antagonism, methysergide, cyproheptadine, pizotifen	Not evaluated in the context of TBI-elicited behavioral changes, though antagonism of the receptor may be prophylactic in the treatment of migraines, whereas other 5-HT_2_ receptor family antagonists like ketanserin do not exert an anti-headache effect [98].
5-HT_2C_	Male and Female 11 individuals	Depression	-	Not evaluated in the context of TBI-elicited behavioral changes, though increased RNA-editing of the transcripts encoding the receptor have been observed in individuals with major depression [99]. Individuals with major depressive disorder exhibited significantly increased expression of the AC’C-edited mRNA and a significantly decreased expression of the A-edited isoform compared to non-depressed individuals.
5-HT_3_	Male and Female, 321 patients enrolled	Substance abuse	Antagonist, ondansetron	Not evaluated in the context of TBI-elicited behavioral changes, though antagonism of the receptor in patients with early-onset alcoholism led to less alcohol consumption as compared to placebo by ameliorating underlying serotonergic abnormalities [100].
5-HT_4_	Male and Female 38 individuals	-	Antagonism	Not evaluated in the context of TBI-elicited behavioral changes, though receptor levels were increased in the frontal cortex and caudate nucleus of depressed victims of suicide compared to controls [101].
5-HT_5A_	Male and Female 502 individuals	Schizophrenia	-	Not evaluated in the context of TBI-elicited behavioral changes, though the Pro15Ser substitution in the receptor is highly associated with the development or progression of schizophrenia symptoms [102].
5-HT_6_	Male and Female	Alzheimer’s Disease	Antagonism, Latrepirdine, idalopirdine, interpirdine, donepezil	Not evaluated in the context of TBI-elicited behavioral changes, though the effects of pharmacological modulation of the receptor on primary cognitive endpoints associated with Alzheimer’s disease have been evaluated in Phase III clinical trials [103]. All trials have failed to demonstrate clinical efficacy and slow progression of dementia symptoms.
5-HT_7_	Male and Female 137 individuals	Schizophrenia and Bipolar affective disorder	-	Not evaluated in the context of TBI-elicited behavioral changes, though two naturally occurring receptor variants were investigated in individuals with schizophrenia and bipolar affective disorder [104], with the data not supporting a role for the receptor in either disorder.

### 4.4. SERT, Cessation of 5-HT Signaling, and SSRIs

Although a significant body of preclinical and clinical research has connected the 5-HT system to the development of psychiatric disorders like depression and anxiety [105,106,107] and increasingly powerful research technologies offering unparalleled insight into the function of the 5-HT system, few insights have been successfully translated into therapies. While our understanding of the nature of the 5-HT system has advanced rapidly in recent years, our ability to functionally modulate the 5-HT system and improve the quality of life for individuals afflicted with emergent psychiatric complications following TBI is extremely limited. For decades, clinicians have relied on 5-HT reuptake inhibitors (SSRIs) to treat depression and anxiety symptoms [55,108,109,110,111]. Following the development of SSRIs as antidepressants in the 1990s, it was widely accepted that changes in the central 5-HT system were causal in generating depressive symptoms [55], although this hypothesis has come under enhanced scrutiny in recent years [112]. As depicted in Figure 3A, SSRIs unilaterally block the uptake of 5-HT through the 5-HT transporter (SERT) into the presynaptic neuron after vesicular release into the synaptic cleft [113], thereby altering the availability of 5-HT to bind to postsynaptic 5-HT receptors [114]. Although SSRIs rapidly increase the abundance of 5-HT in the synapse following administration, the antidepressant effects of SSRIs often take 4–8 weeks to manifest [105]. The temporal discrepancy between the immediate pharmacodynamic action of SSRIs and the latency to perceive changes in mood or affect illustrates that our understanding of the mechanisms underlying the effects of these drugs is, at best, incomplete.

In the context of TBI, studies in humans evaluating the application of SSRIs are inconclusive. Some reports indicate that SSRIs are both well tolerated and efficacious in individuals with TBI (though small sample sizes precluded the accurate determination of predictors of response), while others suggest that SSRIs do not reach the desired efficacy in the context of neurotrauma [19,115,116]. Regardless, antidepressants like SSRIs are still considered to be the best practice for attenuating depressive symptoms following TBI and are routinely prescribed in both in-patient and out-patient scenarios [117], and preclinical research seeking to elucidate the effects of TBI on the 5-HT synapse are ongoing (Figure 3B).

**Figure 3 brainsci-14-00051-f003:**
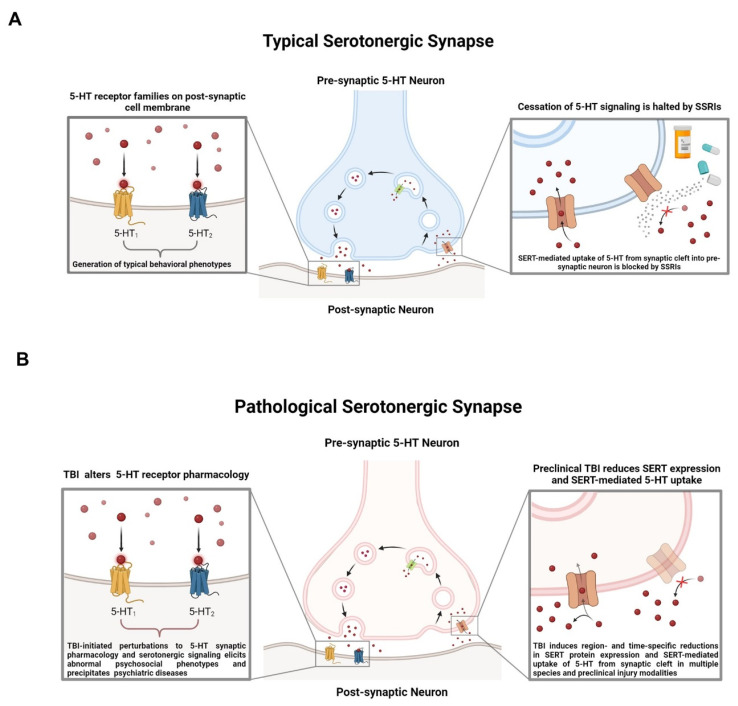
(**A**) Serotonin (5-HT) is packaged into vesicles through vesicular monoamine transporters, then released into the synaptic cleft. In the synapse, 5-HT binds to the 5-HT receptor families present on the membrane on the post-synaptic neuron, eliciting receptor family-specific effects. Cessation of 5-HT signaling occurs when 5-HT is transported back into the pre-synaptic neuron by the 5-HT transporter (SERT). (**B**) TBI disrupts basal signaling through 5-HT receptor subtypes and reduces SERT expression and 5-HT uptake capacity in a region- and time-specific fashion across preclinical rodent models for TBI [118,119].

## 5. Emerging Complexity of the 5-HT System

### 5.1. Complexity of 5-HT Neuron Architectures and Relevance to Traumatic Brain Injury

Though a profound latency in therapeutic effect following immediate SERT blockade may simply be a factor inherent to the pharmacology of SSRIs, it is more likely that a complex and as-of-yet unresolved mechanism underlies SSRI-induced changes to the regulation of serotonergic signaling. This is particularly true in the context of neurotrauma. While imaging technologies and post-mortem studies of human tissue have been instrumental in elucidating important anatomical and functional features of both healthy and injured brains, animal models have enabled increasingly precise interrogation of neural systems at a resolution not currently possible in live humans. For example, next-generation sequencing technologies have granted neuroscientists an unprecedented window into the structure and sub-structure of distinct neuron populations within the murine brain. Recently, spatially resolved transcriptional characterization of the murine brain depicted a greater cell-type diversity in the midbrain than in any other brain region [120], and it is likely this extraordinary heterogeneity enables 5-HT neuron populations to participate in relatively disparate molecular, physiologic, and behavioral processes.

The immense cell-type diversity within the mammalian midbrain may be both advantageous and disadvantageous in the context of TBI. Firstly, the characterization and subsequent pharmacological modulation of the transcriptionally distinct cell populations in the midbrain may permit discovery of therapeutic targets that are highly specific. Secondly, SSRI-elicited alterations in serotonergic regulation and synaptic activity within diverse midbrain cell populations may be relevant in understanding the therapeutic latency of SSRIs and the discrepancies in their efficacy. Finally, and most critically, the midbrain (and midbrain cell populations) is subject to the greatest maximal strain during the acceleration/deceleration events associated with concussions, which may dramatically impact subtle structural, functional, or transcriptional differences between midbrain cell populations. The next generation of medications for depression and the first treatments for TBI-elicited psychiatric disorders will likely emerge from studies that have integrated these factors into the experimental design.

### 5.2. Anatomy, Hodology, and Function of Ascending and Descending 5-HT Neuron Architectures

Midbrain serotonergic neuron populations are extraordinarily heterogenous and exhibit distinct anatomical organization, synaptic hodologies, and transcriptional programs. In mammalian brains, serotonergic neurons emerge during development within the rhombencephalon and later extend into the mesencephalon during processes of gross anatomical reorganization like the folding of the neural tube or cellular migration to the midline raphe nuclei (RN) where the majority of 5-HT neuron populations are stratified anatomically in adulthood [121,122,123,124,125]. The activity and functional organization of serotonergic neurons in the dorsal and median zones of the raphe nuclei have specifically been implicated in an array of emergent mental states, including mood, anxiety, attention, aggression, social status, and learning, in both humans and rodents [46,47,48,126], and, therefore, may be relevant in understanding the generation of emergent mental states following TBI. Though the anatomical groupings of dorsal and median raphe neurons overlap in the brainstem in humans (dorsal raphe; groups B6, B7|median raphe; groups B5, B8), and the primary rostral and caudal groupings are composed of 5-HT neuron subpopulations that share a single anatomical locale in mice, the DR and MR 5-HT networks differ substantially in hodology and developmental origin, and in the governance and generation of behaviors [46,121,127]. In mice, both the DR and MR interface with projections from common brain regions yet propagate signals to predominantly disjoint regions in the forebrain, with DR projections targeting primarily lateral regions and MR projections innervating regions in the midline [46,128,129,130,131,132,133,134].

The functional organization of both the DR and the MR can be even further subdivided, as seminal work has demonstrated significant evidence for subpopulation-specific projection patterns that represent intra-region heterogeneity in both the DR and MR [135,136]. How TBI impacts the highly complex functional organization of the distinct 5-HT neuron subpopulations in the DR and MR is not currently understood. As the RN is preferentially impacted by the acceleration/deceleration events associated with concussions, a comprehensive functional delineation of the anatomically distinct RN 5-HT neurons and their projection architectures is needed to characterize TBI-elicited aberrations in behaviors that are contingent on the functioning of RN 5-HT neurons. Coincidentally, the systems of 5-HT neurons in both the DR and MR subregions of the RN have been exhaustively characterized.

The networks of DR serotonergic neurons that project from the midbrain to the forebrain have been examined in great depth [135]. DR 5-HT neurons can be stratified into two distinct but parallel subsystems that are implicated in reward, punishment, coping, and anxiety behaviors [135]. Retrograde tracing studies revealed that the spatial distribution of DR 5-HT neurons corresponds to specific projection targets in the forebrain [135]. Specifically, 5-HT neurons that project from the DR to subcortical regions like the central amygdala (CeA) are localized to the dorsal zone of the DR, and 5-HT neurons that project to cortical regions like the orbital frontal cortex (OFC) are localized to the ventral DR [135]. Additionally, DR 5-HT neurons comprising cortical projections co-express vesicular glutamate transporter 3 (Vglut3, *Slc17a8*), and consequently also signal using the excitatory neurotransmitter glutamate [135]. Activation of these DR serotonergic neurons influences behavior in a subregion and projection target-specific manner, with both paralleled and complementary behavioral consequences [135]. For example, CeA-projecting DR 5-HT neurons promote anxiety-like phenotypes when activated, but activity of OFC projections does not [135]. Finally, stimulation of OFC-projecting DR 5-HT neurons enhances active coping, whereas modulation of CeA-projecting DR 5-HT neurons has no effect on coping strategies [135].

Within the system of MR serotonergic neurons, similar functional and hodological distinctions between subregion-specific populations of 5-HT neurons in the MR are reported [136]. Importantly, not all MR serotonergic populations express the sets of genes canonically associated with 5-HT signaling. Specifically, a subpopulation of *Pet-1* lineage MR serotonergic neurons that emerge from the same zone of the hindbrain, rhombomere 2 (*R2*), as 5-HT neurons that express tryptophan hydroxylase 2 (*Tph2*), which encodes the rate-limiting enzyme for 5-HT synthesis, instead express Vglut3 (*Slc17a8*) [136]. Additionally, neurochemically distinct populations of r2-*Pet-1^Vglut3-^*^high^ neurons project to and terminate in distinct brain regions, in contrast to *Pet-1^Tph2-^*^high^ populations of R2-derived 5-HT neurons [136]. Like Vglut3-high populations of DR neurons, r2-*Pet-1^Vglut3-^*^high^ neurons also preferentially project to cortical regions and modulate the activity of cortical neuron populations through the release of glutamate [136]. r2-*Pet-1^Tph2-^*^high^ populations innervate midline structures including the suprachiasmatic nucleus, whereas r2-*Pet-1^Vglut3-^*^high^ populations innervate cortical and hippocampal zones [136]. Finally, it is likely that activity of r2-*Pet-1^Tph2-^*^high^ alters circadian rhythm and sensory processing, and that r2-*Pet-1^Vglut3-^*^high^ populations may regulate sensorimotor gating and modulate theta waves in the hippocampus, with effects on memory formation [136]. The tightly regulated anatomical localization, projection architectures, and neurochemistry of DR and MR 5-HT neurons illustrate the subtle complexities of the 5-HT system and simultaneously warrant further investigation in the context of TBI-elicited psychiatric phenomena.

### 5.3. 5-HT Neurons Organize into Transcriptionally Distinct Subpopulations

It is perhaps unsurprising that an increasingly rich functional diversity within the serotonergic system has been revealed, as the resolution with which the DR and MR serotonergic systems can be characterized has increased considerably following the advent of spatial transcriptomics and other innovative next-generation technologies. The marked heterogeneity within these neural systems persists from the scale of the anatomical to the neurochemical, cellular, and transcriptional scales. The remarkable specificity in projection architecture and synaptic function discussed above necessarily emerges from tightly regulated differences in the transcriptome at the subpopulation level. Indeed, recent seminal works have characterized at least 11 transcriptionally distinct subclusters of 5-HT neurons within the DR and MR that employ unique transcriptional regimes to maintain population-specific hodology and functionality [46,137], though the actual number may be greater than 14 distinct populations. While the classical conceptualization of serotonergic neurons requires the basal production of 5-HT and the synaptic release of 5-HT, a more accurate conceptualization of serotonergic populations likely encompasses a much wider range of unique phenotypic or transcriptional features. To integrate the functional and transcriptional diversity discussed above into the framework that defines serotonergic neural systems, one might ask the question: “What features are required for a neuron to be designated as serotonergic?” Anatomical localization alone does not define serotonergic populations, as neurons from the same rhombomere-2 *Pet-1* derived lineage employ disjoint sets of neurotransmitters [136]. The absence of basal 5-HT production does not preclude inclusion into the serotonergic framework, as some RN neurons signal predominantly or exclusively through the release of glutamate [135,136]. To better understand the molecular mechanisms that catalyze mental health disorders with complex etiologies like MDD, these observations become increasingly relevant, especially in the context of TBI. While an extended conceptualization of the 5-HT system will allow scientists to make informed, mechanism-derived decisions in drug discovery, it might also challenge the established ideas about how these neural systems function and how perturbations to these systems generate psychiatric phenomena like depression and anxiety.

### 5.4. Therapeutic Perspectives

While the aforementioned recent research efforts have yielded valuable insight into the basal function and organization of 5-HT neuron populations within the RN, our current understanding of how the 5-HT system is affected by various forms of neurotrauma is limited. It is known that closed-head neurotrauma increases 5-HT levels within the brainstem [58] and blast-induced TBI is reported to acutely increase mRNA levels of *Tph2* in the DRN in a time-dependent manner [52,59,138]. In an open-head preclinical model of TBI, it was found that penetrative head injury disrupts 5-HT neuron tracks in the cerebral cortex, with significant regrowth occurring over time [139,140]. 5-HT neuron projections have the unique capacity to undergo axonal recovery and dendritic arborization despite glial scars, but new axonal projections assume altered hodologies and spatial orientation [139,140], an effect with unknown implications for receptor expression/densities, neuronal network function, and behavioral phenotypes. Though modern drug discovery efforts focused towards attenuating the 5-HT-dependent neurobehavioral disturbances elicited by TBI have not yet yielded new FDA-approved medications, the effects of pharmacologic modulation on the activity of the family of 5-HT receptors have been an area of significant experimental interest. A plethora of 5-HT receptor binding chemical ligands that modulate the activity of the 5-HT system have been reported, with distinct effects on behavior [61]. In addressing the inefficacy of SSRIs in treating TBI-induced neuropsychiatric disturbances, selectively targeting and modulating the activity of specific 5-HT neuron subpopulations within the RN represents a promising target for therapeutic development strategies. This is because there are discrete correspondences between the anatomical localization of 5-HT RN neuron subpopulations, the constituent gene expression profiles maintained within these subpopulations, the afferent regions the subpopulations project to, and the resulting behaviors that arise from 5-HT signaling to downstream regions.

Closed-head neurotrauma acts to initiate cascades of transcriptional adaptations that are known to be cell-type specific [141]. The dynamical differential expression of gene sets within distinct subpopulations of 5-HT DRN neurons is not only critical for the maintenance of cell-type identity but is also imperative for sustaining proper functional tone of these neurons as it pertains to basal signaling to associated projection regions. A wide body of literature demonstrates that TBI induces dynamic changes to gene expression at various timepoints after injury, and it is perhaps unsurprising that the RN, a region preferentially impacted by closed-head neurotrauma, would not be spared from TBI-induced changes to transcriptional regulation. Given that dynamic transcriptional activity precedes changes in protein expression and functional adaptations within neuronal populations [142], it is relevant to understand the mechanisms that govern transcriptional control of 5-HT RN neurons in a subpopulation-specific manner in the context of TBI. Like other somatic cells, neurons rely on the expression of transcription factors (TFs) and epigenetic modifiers (EMs) to exert control over transcriptional regimes that subsequently govern cellular function [143]. 5-HT neurons within the RN maintain transcriptional heterogeneity across anatomical localizations, and the abundance or activity of transcriptional control processes, i.e., TFs and EMs, is expected to be unique to neuronal subpopulations within the RN [137]. There is evidence for TF dysregulation in the DRN in individuals with MDD [144], suggesting that mood disorders with an origin in serotonergic signaling do indeed have a more complex etiology than reduced synaptic levels of 5-HT, necessitating development of drug classes beyond SSRIs.

## 6. Conclusions and Summary

Despite historical and ongoing interrogation in both preclinical and clinical settings, there are no FDA-approved pharmacotherapies to attenuate TBI-induced neuropsychiatric disturbances. The paucity of specific, mechanism-derived therapies to ameliorate the psychiatric complications associated with TBI has led to a tremendous economic burden born by the public at large, with equally monumental social-emotional ramifications at the level of the individual. Though decades of research provide empirical evidence for the benefit of pharmacologic manipulation of 5-HT at a receptor-specific level in the treatment of psychiatric disorders, the prescription of SSRIs remains the best practice. Importantly, comparatively few FDA-approved medications that selectively target specific 5-HT receptors are applied in the clinic to attenuate psychiatric disturbances. Concurrently, a significant research effort has delineated the vast anatomical, transcriptional, and neurochemical subtleties of the 5-HT system at large, in the absence of neurotrauma. Loss of precise transcriptional control exercised within serotonergic neurons through the concerted activity of transcriptional control mechanisms like TFs and EMs following TBI likely precipitates protracted changes in 5-HT neuron identity, function, and neurotransmission that underly various comorbidities. Future drug discovery paradigms for TBI-elicited psychiatric disturbances may be accelerated by integrating empirical evidence of serotonergic neuron transcriptional reprogramming following neurotrauma into appropriate conceptual frameworks.

## Figures and Tables

**Figure 1 brainsci-14-00051-f001:**
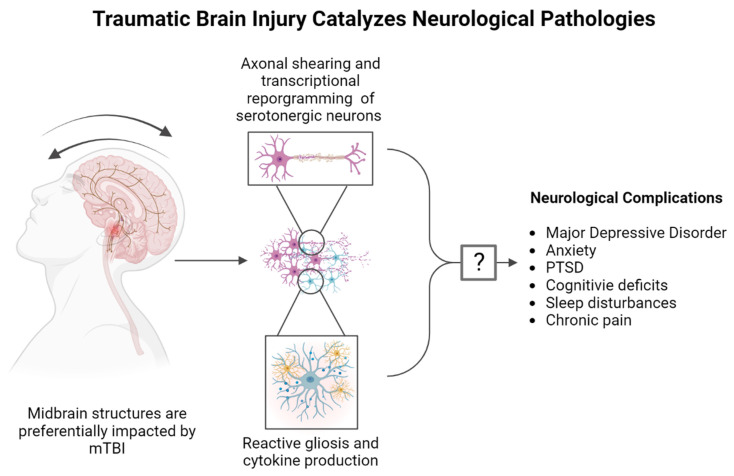
Mild traumatic brain injury (mTBI) has been found to preferentially affect the midbrain due to the innate physiology of the region coupled with the biomechanics of various forms of head injury, including repeated head impacts (RHI). The midbrain contains a multitude of neuronal cell bodies responsible for serotonin (5-HT), dopaminergic and glutamatergic neurotransmission with projections that ascend to distinct brain regions implicated in the regulation of diverse behavioral phenotypes and descend to the periphery. Disruption of the cellular milieu within the midbrain may be a catalyst in the formation of neuropsychiatric disturbances and chronic pain that often arise from diverse forms of neurotrauma.

**Figure 2 brainsci-14-00051-f002:**
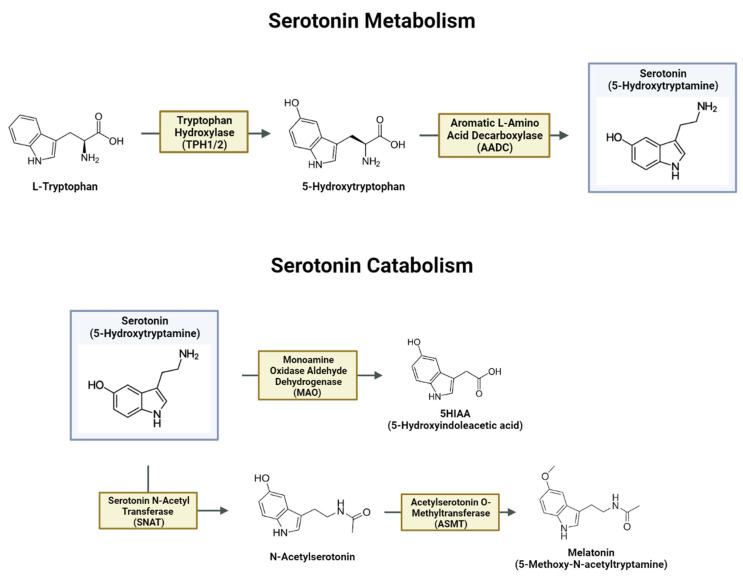
L-Tryptophan is metabolized into 5-Hydroxytryptophan by rate-limiting enzyme tryptophan hydroxylase-2 (Tph2), and then into 5-HT by aromatic l-amino acid decarboxylases (AADC). Major 5-HT catabolism pathways yield the primary metabolite 5-hydroxylindoleactic acid (5HIAA) and melatonin, which is bioactive and important in the regulation of sleep.

## Data Availability

Not applicable.

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
