# Peer review of "Strain in the Midbrain: Impact of Traumatic Brain Injury on the Central Serotonin System"

_brainsci, 2024, doi:10.3390/brainsci14010051_

Round 1
Reviewer 1 Report
Comments and Suggestions for Authors
The manuscript by O’Connell and colleagues addresses an important area within the field of traumatic brain injury. Understanding the pathophysiological disruptions to important monoaminergic systems could yield some important insights. Critically as so many therapeutics exist to modulate serotonergic signalling, repurposing of medications could feasibly occur in the short term to alleviate the suffering of those individuals with long term neurobehavioral deficits induced by TBI. However, there manuscript requires reorganization and modification of the tables to really highlight their findings. Moreover, it appears as thought the review is more narrative in nature and does not follow PRISMA guidelines. More detailed comments are outlined below.
· Throughout the manuscript there needs to be a specific focus. It would be best if the authors make a definitive focus, will the review evaluate all changes in subjects that have ever had a TBI (independent of severity) across a specified timeline, acute subacute chronic time points post injury, or is review to focus only on studies relevant to long-term illnesses that emerge in the months to years following recovery of the initial injury?
· The introduction should be reframed with the main purpose of the review stated up front (last paragraph of current introduction). Then the authors should detail what the review will evaluate and provide references to reviews that cover topics not covered with the manuscript. All though much the information in the introduction is important, it is not entirely useful and actually detracts from the primary goal of the review.
· Methods are missing. Please ensure that PRISMA guidelines are followed where possible and include the flowchart used. This is particularly important for the clinical studies. Indicate the type of review, scoping, systemic or narrative. Which platforms were used for the search, keywords, search periods etc. This is now standard practice for all reviews. Narrative reviews while of interest are generally perceived as a biased view point, but as long as this limitation is acknowledged, this is acceptable.
· Be conscientious throughout in reporting findings. For clinical studies, report demographics of the patient population, severity of TBI, (mild, moderate or severe), treatment outcomes with specific changes in scales used. Indicate whether self-report or clinical ratings were used to identify neurobehavioral deficits in those with neurotrauma. Please also indicate that and the timing of behavioral deficit assessment or intervention post TBI. For preclinical studies, be specific in the model used, and the measures used, do not anthropomorphise the results.
· In section 3, these findings are very interesting, but very list like, and disjointed in the current presentation. Throughout the field we are all obsessed with timing. In reading this I can see where you have delineated important time points in 5-HT metabolism. Use the timeline to your advantage - break this section down accordingly. What changes, if any, occur at the initial injury time-point, the sub-acute time points and chronic (7 days) or long term (1 month or 1 year etc). This presentation would be very valuable and you can point to the key periods as the times at which interventions would be optimal. This section lacks some clarity. It is uncertain whether the changes reported occur in the context of TBI.
· Related to section 3, revise your tables in a TBI centric fashion and break it down according to preclinical or clinical studies. The field can look up agonists and antagonists easily, your contribution should be to tell us more about 5-HT in the context of TBI at a glance. You should include the following columns, for each receptor indicate the subjects of interest (species, age etc. the usual demographics), the TBI diagnosis (clinical) or model used (preclinical), outcomes should be specific for preclinical studies not just a list of effects on behavior. If pharmacology was used to reverse, or exacerbate phenotypes please indicate accordingly.
· In section 3.4, serotonin reuptake inhibitors are generally referred to as selective serotonin reuptake inhibitors (SSRIs). Much of this section is not really of relevance, merely a conversation regarding SSRIs and where they might work, but there is no recommendation or evidence to suggest that would be of any greater benefit to those that develop major depressive disorder of anxiety following a TBI. Moreover, it is suggested that these medications may only be of benefit to moderate or severe depression. However, it should be noted that often the most clinically ill are those that do not exhibit a response to these medications.
· Although figure 2 and 3 are accurate, they do not enhance the value of the manuscript itself. The 5-tryptophan metabolism and the actions of SSRIs at the synapse of serotonergic neurons are well documented pharmacology that is quite redundant here. It would be more helpful to denote changes in these synapses post TBI, perhaps even in mild, moderate and a severe case. Are there documented differences?
· Section 4.1 is not useful – no particular recommendation of point to this paragraph.
· Section 4.2 is the most exciting section. The sub-title itself should be its own heading, forget about the expanding framework title.
· I really do appreciate the delineation along DR and MR, but would be really excited to hear if you had found any changes in the descending inhibition of pain post TBI. This would be particularly of relevance in those patients with pain following a TBI.
· It would be good to have accompanying tables for sections 4.2 and 4.3, following those recommendations made for Table 1.
· Although I appreciate that the focus is in serotonin, you may wish to acknowledge the kynurenine pathway and its metabolites briefly at the end. There are some interesting findings with this pathway post TBI.
Reviewer 2 Report
Comments and Suggestions for Authors
The paper entitled “Strain in the Midbrain: Impact of Traumatic Brain Injury on the Central Serotonin System” by Christopher J. O’Connell and the co-authors is an excellent contribution in unveiling the role of TBI in dysregulating serotonergic neuronal functions and neurotransmission.
Overall, the paper is nicely designed with tables and figures.
A full explanation has been added in each section.
The figures are precisely designed and integrated in a nice way.
The conclusion is fine, but for future studies, the author may add specific research gaps in current understanding regarding the pathogenesis of TBI.
The conclusion is fine.
The paper may be minorly revised, and the comments of the other reviewers may be considered.
The individual components (section) of Figures 1 and 3 may be labeled numerically, and the explanation may be given in the legend according to the figures. This will make it easy to browse and understand.
Thank you, indeed it’s a nicely designed study for TBI-associated researchers and scientists.
Round 2
Reviewer 1 Report
Comments and Suggestions for Authors
I appreciate the efforts of the authors and their championing of this topic and therefore accept the limitations of their review process.
The update to Figure 3 is good.
For the tables I would suggest that the information be separated into clinical and non-clinical tables for clarity. For human outcomes don’t just state better outcomes for reference 70. Include more details. The data on migraine is potential very interesting here.
